# Comparative Analysis of Plaque Removal and Wear between Electric–Mechanical and Bioelectric Toothbrushes

**DOI:** 10.3390/bioengineering11050474

**Published:** 2024-05-09

**Authors:** Jihyun Lee, Hyun M. Park, Young Wook Kim

**Affiliations:** 1Department of Periodontology, Ulsan University Hospital, College of Medicine, University of Ulsan, 877 Bangeojinsunhwando-ro, Dong-gu, Ulsan 44033, Republic of Korea; 0733439@uuh.ulsan.kr; 2PAIST (ProxiHealthcare Institute for Science and Technology), ProxiHealthcare Inc., Seoul 08507, Republic of Korea

**Keywords:** bioelectric effect, oral health, plaque formation, plaque removal, tooth wear, gingival index

## Abstract

Effective oral care is important for maintaining a high quality of life. Therefore, plaque control can prevent the development and recurrence of periodontitis. Brushing with a toothbrush and toothpaste is a common way to remove plaque; however, excessive brushing or brushing with abrasive toothpaste can cause wear and tear on the dental crown. Hence, we aimed to quantitatively compare the plaque-removal efficiency and tooth wear of toothbrushes using the bioelectric effect (BE) with those of electric–mechanical toothbrushes. To generate the BE signal, an electronic circuit was developed and embedded in a toothbrush. Further, typodonts were coated with cultured artificial plaques and placed in a brushing simulator. A toothpaste slurry was applied, and the typodonts were eluted with tap water after brushing. The plaques of the typodonts were captured, and the images were quantified. For the tooth wear experiment, polymethyl methacrylate disk resin blocks were brushed twice a day, and the thickness of the samples was measured. Subsequently, statistical differences between the experimental toothbrushes and typical toothbrushes were analyzed. The BE toothbrush had a higher plaque-removal efficiency and could minimize tooth wear. This study suggests that the application of BE may be a new solution for oral care.

## 1. Introduction

Oral health is crucial for improving the quality of life [1]. However, according to the World Health Organization (WHO), 3.5 billion people worldwide suffer from oral diseases [2,3], with plaque formation being a primary cause [4,5]. Plaque is a group of bacterial biofilms that has been identified as a major cause of periodontitis or periodontal disease, which is directly linked to various health conditions, such as stroke, diabetes, cardiovascular disease, and Alzheimer’s disease [6,7,8,9]. Plaque comprises over 1000 species of bacterial strains with an extracellular matrix (ECM) to protect against external stimuli including antibiotics [4,5]. Due to the complex structure of plaque, traditional antibiotic treatment methods are not efficient since the drug is not able to diffuse into the ECM. Consequently, effective plaque removal by physical brushing is important in both public and private healthcare for the maintenance of good-quality oral health.

Biofilms are composed of a protective layer of polysaccharides and electrically polarized multispecies of bacterial cells [10,11]. Since antibiotics can hardly penetrate the ECM, biofilms are extremely persistent, as they typically require 500–5000 times more concentration of antibiotics than planktonic bacteria [12,13,14]. Hence, biofilm-associated infection, including gingivitis, usually requires physical surgery and intensive chemical washing that includes significant pain and a high cost. Therefore, there is a great interest in developing less-invasive therapy for biofilm infections.

The application of an external electrical current can be an alternative method for oral plaque infection based on the principles of the bioelectric effect (BE) [15,16]. Biofilms are composed of electrically charged molecules and chemical bonding on the surface, such as peptide and hydrogen bonding. Thus, when an electric field is applied to biofilms, the surface-charged molecules can be influenced by the electrostatic force via the Coulomb force law. This electric effect on biofilms can induce enhanced permeability of the ECM via alternating current (AC), and the inhibition of bonding strength maintenance of biofilms as the field can distort electrolyte equilibrium conditions, especially the concentration of hydrogen via direct current (DC) electric stimuli. These AC and DC electric fields eventually weaken the structure of biofilms and make biofilms more susceptible to low doses of antibiotics, as well as increase the efficacy of biofilm removal, known as the BE [15,16,17,18,19,20]. Our group has developed a combinatorial BE to reduce electric power consumption as well as produced dedicated biocompatibility toward healthcare applications.

When external electric power is applied in a water-based solution, the prevention of carcinogenic radical ion generation, such as hypochlorite (ClO^−^), hypochlorous acid (HOCl^−^), and hydroxide (OH^−^), is critical to healthcare applications [15,20]. We carried out a quantitative analysis of the electrochemical impact due to the BE and demonstrated the biocompatibility as shown in non-electrochemical condition changes due to the electric current applications with intensive clinical studies previously [15,16,17,18,19,20]. Hence, the BE can be an additional effective method for biofilm removal in healthcare industries [17,18,19,20]. Recently, our group developed a toothbrush with the BE, which was shown to reduce inflammation (gingival index) in clinical trials [12,20] due to the significant improvement in plaque removal. We also demonstrated in an in vitro brushing simulator study that BE toothbrushes showed significantly more efficiency at removing plaque than conventional toothbrushes [21].

In the 1950s and 1960s, electric–mechanical toothbrushes with high plaque-removal efficiencies were developed [22,23]. These toothbrushes have a higher plaque-removal rate than regular toothbrushes because the electric motor utilizes rotational and planar vibration [24,25,26]; however, they cause tooth wear [27]. Moreover, electric–mechanical toothbrushes can accelerate tooth wear when used in conjunction with charcoal toothpaste [28]. Dental abrasion, a type of non-carious cervical lesion (NCCL), is defined as tooth wear caused by excessive brushing or brushing with an abrasive toothpaste [29] (Figure 1). Tooth wear is slow, progressive, and irreversible; consequently, exposed cervical dentin can cause dentin hypersensitivity, a sharp pain that may require dental treatment [30]. The severity and prevalence of NCCL are likely to increase with age [31]. Therefore, preventive dental care is important, and new toothbrush devices that can prevent NCCL at a young age are required.

In this study, we aimed to perform a quantitative investigation of the plaque-removal efficiency and surface wear effect focused on the BE toothbrush and electric–mechanical toothbrushes, which have high plaque-removal efficiency but are disadvantaged in terms of tooth wear. Analysis of the material wear due to the surface friction of movement in toothbrush requires precision control of brush parameters, including applied pressure, brushing direction, and repetition, along with measurement of tooth wear. These are not adaptable parameters for human-involved testing. Clinical trials are not appropriate for quantitative evaluation owing to inter-individual variability and the difficulty of monitoring; however, in vitro simulators have been used to increase the reproducibility of the experiments, including toothbrush efficacy studies [27,32,33,34]. The simulator was integrated with a precise displacement controller, corresponding to the applied pressure of the toothbrush on tooth-mimicked material. It selected polymethyl methacrylate (PMMA) for use as a simulated tooth based on previous literature [35]. The brushing direction and repetitions are set by the linear actuator and step motors of the simulator [21]. The thickness measurement of the PMMA wear was conducted using a micrometer with three repetitions. The simulator demonstrated excellent correlation with standard pressure sensors and displacement [21]. We conducted four types of toothbrush testing conditions (Control: conventional toothbrushes, Negative Control: developmental toothbrushes without the BE, Experimental Group: with the BE, and Experimental group: electric–mechanical toothbrushes) using the in vitro toothbrush simulator. We used Streptococcus mutan-based plaque incubated in a growth medium as the standard model of oral biofilms [36]. Based on the experimental results, we investigated the degree of plaque removal and surface wear effect of the BE toothbrush in comparison to the traditional electric–mechanical device. 

## 2. Materials and Methods

### 2.1. Tested Toothbrushes and the Design of BE Toothbrush

The major purpose of this study is to investigate the surface wear effects of different plaque removal methods. All toothbrushes include bristle as a physical plaque removal mechanism. The key variants are whether the BE or electro-mechanical motor-driven force is applied. To set adequate conditions for testing, we chose four commercially available toothbrushes: (1) Oral-B^®^ ultra-fine, (2) Developed non-BE, (3) Developed with BE, and (4) Oral-B iO3 electric–mechanical devices. The conditions of the proposed toothbrushes were specified to investigate the effect of bristles (typical toothbrush versus the proposed toothbrush) on plaque removal. Details of information are shown in Table 1 and Figure 2. 

We compared the plaque-removal efficiency and PMMA wear of three toothbrushes with those of a BE toothbrush. 

The BE is a combination of electrical signals, specifically a 0.7 V sinusoidal signal at 10 MHz with a DC offset of 0.7 V, as shown in Table 2 and Figure 3 [12,20,21]. The BE signal frequency was selected to maximize the permeability of biofilms through the ECM and demonstrated 400 times enhancement, reported in a previous study [15]. The DC offset was applied to optimize the weakening of surface bonding on biofilms as well as set below an electrolysis threshold of 0.82 V [15]. This is critical to ensure biocompatibility [15,20]. Our previous studies revealed that this signal did not produce byproducts of electrochemical radicals due to the decomposition of water [15,20,21] but showed significant biofilm reduction [15,20]. The BE toothbrush is integrated with a rechargeable Lithium-ion battery (3.7 V) to operate electronics. Since the BE is characterized by the consideration of biosafety, the current from the BE signal was determined to be safe [20,21], referring to the reported biocompatible current range, under 1000 micro-ampere in pH7 saline condition [37]. The BE toothbrush is internationally certified for home appliance safety, including by the US-FDA, EU-CE, and Korea. Thus, it is safe to use. 

An electronic circuit was developed to generate the BE signal, as shown in Figure 4 [12,20,21]. Subsequently, this circuitry was embedded in a toothbrush. A stainless-steel electrode was selected because of its corrosion resistance and conductivity, which are essential for supplying an electric field. The signal for the BE was output through the two electrodes.

### 2.2. The Brushing Simulator

The brushing simulator used in this study was developed in our previous study and is shown in Figure 5 [21]. This system was focused on accurate brushing conditions, that is, pressure, direction, displacement, and repetition, with which patients involved in clinical trials are challenged. The main components were a linear actuator, stepper motor, motor driver, piezoelectric pressure sensor, and an artificial tooth construct. An Arduino Uno (Arduino, Italy) was used for simulation control and data acquisition (DAQ). The system was calibrated and validated in performance [21].

### 2.3. Plaque Culture

For the plaque removal efficacy testing, Streptococcus mutans (KCTC 3065, Korean Collection for Type Cultures (KCTC), Jeongeup, Republic of Korea) was chosen with consideration as a simulated plaque condition [37]. The strain was cultured in a growth medium (LB Broth, Ambrothia Inc., Daejeon, Republic of Korea) at 37 °C for 48 h to provide sufficient time for maturation.

### 2.4. Experiment

In this study, experiments were conducted to investigate two aspects of toothbrushes: plaque removal and wear (surface friction), as shown in Figure 6.

#### 2.4.1. Plaque Removal Experiment

A plaque-removal experiment was designed to investigate the efficacy of plaque removal using each toothbrush. We utilized typodonts to compare the plaque removal efficacy in the consistent condition. The typodonts were coated with cultured *S. mutans* plaques. Green marker spray (OccludeTM; Pascal, Bellevue, WA, USA) was used to visualize the plaque [34]. Subsequently, the coated typodonts were placed in a brushing simulator. 

The brushing pressure condition on the surface, referred to by ISO/TR 14569-1, specifies the load a toothbrush exerts from 0.5 to 2.5 N [38]. The force with which the toothbrush pressed on the teeth was set to 1.5 N as the average value from the suggestion and the flat surface of molars in the typodonts was brushed to minimize the variant of cleaning due to tilted surface [34]. The brushing speed was 120 strokes/min, as suggested in the literature [36]. Two minutes is generally the recommended duration for toothbrushing [37]. The total time for brushing was set as 30 s since the molar region is approximately taken as one-fourth of the entire teeth. To mimic real-world toothbrushing conditions, a toothpaste slurry (toothpaste/saline ratio of 1:3) was applied [21]. 

After brushing experiments, typodonts were eluted with tap water for 30 s. This process removed isolated plaques for quantification. Our preliminary testing revealed that this process prevents unintended plaque elution [21]. The plaques of the typodonts were captured using a DSLR (Digital Single Lens Reflex, Nikon D3100, Tokyo, Japan) camera under controlled lighting conditions before and after the experiment. The image was converted by a binary (black and white) function and quantified the percentage of the surface coverage area using a standard image analysis software (Image J 1.4.4., National Institutes of Health, Bethesda, MD, USA).

#### 2.4.2. Wear Experiment

A surface wear experiment was designed to investigate the quantitative measurement of friction caused by each toothbrush. According to the standard condition of brushing, ISO/TR 14569-1 [38], the vertical force applied to the tooth was between 0.5 and 2.5 N [38]. In this experiment, the pressure applied to the teeth-simulated material was set to 1.5 N, which was selected as the median value. Additionally, the toothbrushing speed was set to 120 strokes/min [38]. As the tooth-mimicked material, PMMA was utilized based on current uses of dentistry thanks to the material composition [36]. A disk resin blocks of the PMMA (10 mm, 10 mm, 20 mm: width, length, height) (A3; Yamahachi, Gamagori, Japan) were used as test specimens. To provide humidity as a mimic tooth, PMMA samples were placed in deionized water at 37 °C for one week [39]. 

Typically, the average person has 28 teeth [40]. A recommended brushing time is 2 min [41], which corresponds to 4.3 s per individual tooth. For the convenience of calculation, assuming there are three sides per cleaning of each tooth, one-side cleaning time can be set to 1.5 s per brush. Teeth were brushed twice a day, which is equivalent to 3 s per day for one side of the tooth. Therefore, one-side brushing for a year is calculated 3 s times 365 days, resulting in approximately 18 min per year. Based on this approach, the testing for PMMA wear has been performed for 18 min to investigate the annual wear effect of each toothbrush. 

The sample dimension of PMMA (10 mm, 10 mm, 20 mm: width, length, height) is smaller than the contact area (13 mm, 23 mm: width, length) of the brush to eliminate the possibility of non-uniform wear. The thickness of the samples was measured 3 times in each sample using a digital micrometer before and after the experiment. Based on the precision control of brushing parameters, each sample was repeated three times, and the average was measured using standard deviation for analysis.

### 2.5. Statistical Analysis

Data are presented as means and standard deviations (SDs). One-way analysis of variance (ANOVA) followed by Dunnett’s post hoc test was used to determine the significance of differences between the experimental toothbrushes and comparison toothbrushes. Paired t-tests were used to compare the differences between toothbrushes with BE (BE-on) and electric–mechanical toothbrushes (MB). All statistical analyses were performed using R-studio version 4.3.1 (Posit, Boston, MA, USA). *p*-values of less than 0.05 were considered significant.

## 3. Results

### 3.1. Experiment of Plaque Removal

The experimental results are shown in Figure 7. No significant difference was observed in the residual plaque between the non-bioelectric effect (BE-off) and CB groups. A previous study reported that no significant difference was observed in the plaque-removal rate according to the shape of the bristle [41]. However, the BE-on and MB treatments resulted in significantly reduced residual plaques. The BE is effective in removing biofilms, such as plaque, which corresponds to the previous finding that the BE toothbrush significantly amplified plaque removal [21]. Electric–mechanical toothbrushes have a higher plaque removal rate than regular toothbrushes [24,25,26]. However, no significant differences were observed between the BE-on and MB groups. The CB, BE-off, BE-on, and MB showed residual plaque percentages of 8.63 ± 0.67%, 7.64 ± 0.14%, 3.97 ± 0.43%, and 3.69 ± 0.09%, respectively.

### 3.2. Experiment with Wear

The measured wear efficacy was shown as the highest value in the MB group, as presented in Table 3. No significant difference was observed between the other groups. Electric–mechanical toothbrushes have been reported to induce greater PMMA wear than regular and BE toothbrushes [27]. No significant difference was observed between the proposed toothbrushes (both BE-off and BE-on) and the CB. 

The MB showed a 138% reduction in the PMMA thickness compared to the BE-on. However, no significant difference was observed in the wear between the BE-off and BE-on groups shown in Figure 8.

## 4. Discussion

This study evaluated the plaque-removal efficiency and PMMA abrasion in different mechanisms of plaque removal using an in vitro brush simulator. The BE utilizes electric force and interference of the ECM in plaque removal that does not depend on the physical cleaning method. A traditional electric–mechanical toothbrush, based on rotational motion powered by an electric motor, has been tested as a comparison of the BE. 

No significant difference in the number of residual *S. mutans* plaques was observed between the CB and BE-off groups. The BE-off does not involve the BE and should be considered a manual toothbrush. No significant difference in plaque-removal efficiency was observed owing to differences in bristles [41]. Similar results were observed in the present study. A significant reduction in residual plaque was observed for the MB and BE-on toothbrushes when compared with that of manual toothbrushes. MBs are more efficient for plaque removal than manual toothbrushes because of the mechanical rotation of their bristles [24,25,26]. We observed a significant reduction in the residual plaque when using BE-on compared with that of a manual toothbrush, which corresponded to the results of previous studies [21].

There is no significant difference in plaque removal efficacy between the MB and BE. This means that electric motor-driven physical plaque removal efficacy can correspond to the effectiveness of the electrostatic force-influenced mechanisms of the BE. Hence, the BE can be considered an alternative method for oral plaque cleaning.

The application of the BE has been demonstrated in the high efficiency of biofilm treatment [17,18,19]. The BE is caused by the propagation of electromagnetic currents and electric charge flow under the induced voltages [14,15,16]. The biocompatibility of the technology was validated via the verification of no biocidal effect, that is, investigation of the in vitro electrochemical condition studies [15,16] and in vivo human clinical studies [12,20]. Applying AC at a specific frequency can improve the porosity of the biofilm structure, especially on the ECM [17,18], whereas applying DC can induce changes in the electrolyte state [15,19]. The experimental conditions consisted of an electrical dielectric using a toothpaste solution containing saline that was used as saliva in several studies [42,43]. Due to these mechanisms of action, the BE toothbrush demonstrated its competitive plaque removal effect compared to the MB and showed higher removal efficiency than the manual toothbrush.

PMMA was used in the in vitro experiments to evaluate wear during brushing. PMMA can be used as an alternative material for wear testing despite the differences between real teeth and PMMA. Although the elastic modulus, hardness, and wear resistance of PMMA are poor compared to those of natural teeth, leading to rapid wear of PMMA block [41], PMMA is reported as a teeth-mimicked material in various dental studies [44,45,46]. Therefore, wear was compared using relative rather than absolute values. No significant difference was observed in the wear rate of PMMA for BE-off and BE-on compared with that of regular toothbrushes. This result demonstrates that the BE does not impact surface wear even if the BE is effective for plaque removal. Microcurrents are known as having no effect on dental enamel [42]. The BE induces a current of 40.7 ± 1.5 μA in saline, which is biocompatible to the human body, as also shown in a previous study [21]. Therefore, in this study, we expected that the BE energy applied to the toothbrush would not affect wear. However, MBs exhibited significant wear because of the strong friction induction due to the electric motor drive physical brushing. The BE-off is the state in which the BE is not applied; therefore, it is equivalent to the manual toothbrush. In addition, the BE-on demonstrates no significant difference was observed in the wear rate between the manual toothbrushes. In contrast, the MB caused significant wear of the PMMA. MBs have been reported to cause wear due to the strong electric rotational brush, and we believe that this study also agreed with the work [27,28].

As the population ages, retaining their natural teeth for longer periods with the prevention of tooth wear becomes increasingly important for good quality of life. The significantly smaller PMMA wear of BE-on compared with that of MB indicates that the application of the BE as a new toothbrush can contribute to the preservation of natural teeth.

Based on the experimental results, we expect that the application of BE will increase plaque-removal efficiency without accelerating the wear of permanent teeth. The BE toothbrush not only had a higher plaque-removal efficiency than a regular toothbrush but also had a plaque-removal rate equivalent to that of an MB. The wear due to the BE was observed to be comparable to that of a manual toothbrush but significantly lower than that of the MB. These results suggest that BE toothbrushes have high plaque-removal efficiency and low wear rates. In addition, clinical studies have seen toothbrushes significantly improve the gingival index [12,20]. Therefore, BE toothbrushes may provide a new way of plaque removal for oral healthcare. The BE toothbrush can also be an economic tool for oral health maintenance based on the same cost and lifetime with ordinary uses compared to the traditional electric–mechanical device.

Experiments using in vitro simulators are not only more reproducible than clinical trials but also enable precisely controlled quantitative experiments [21]. In this study, we quantitatively evaluated the plaque-removal efficiency and wear of various toothbrushes. In our previous clinical study, BE-treated toothbrushes were effective in improving gum inflammation due to the high efficacy of plaque removal [12,20]. 

Plaque is a root cause of various oral inflammation, including of the teeth, gum, and tongue [47]. Especially when plaque travels through the blood stream, inducing gum bleeding symptoms, the bacteria in plaque can cause systematic diseases, such as stroke, cardiovascular disease, and diabetes [6,7,8,9]. For this group of chronic patients, it is more important to maintain good cleaning efficacy of plaque. Moreover, plaques appear in pets [48]. Therefore, we plan to develop various types of oral products, including plaque-related tongue scrapers and oral care products for pets, using the BE. 

This study had some limitations. First, we used constant pressure and brushing speed; thus, changes in toothbrushing patterns in real life are not reflected [49]. Second, we evaluated plaque-removal efficiency in the same typodonts and area; thus, oral structures that vary from person to person are not reflected [50]. Third, wear was evaluated using PMMA as the dental material; however, this may differ from actual teeth [45]. 

Based on this work, extended clinical research should be performed to further apply the BE toothbrush in plaque removal and tooth wear in real environments. 

## 5. Conclusions

In this study, the BE toothbrush shows an effective plaque-removal rate, as well as a reduction in surface wear in comparison to the electric–mechanical device. Hence, it is concluded that the application of BE technology as a toothbrush improved plaque removal efficacy of the manual toothbrush along with minimized surface wear compared to the electric–mechanical brush. Based on these unique advantages of the BE, the toothbrush may be a new and more efficient alternative method, especially focused on preservation of wear with competitive plaque cleaning. For future work, we plan to demonstrate various oral care products integrated with the BE with details of mechanism investigations. 

## Figures and Tables

**Figure 1 bioengineering-11-00474-f001:**
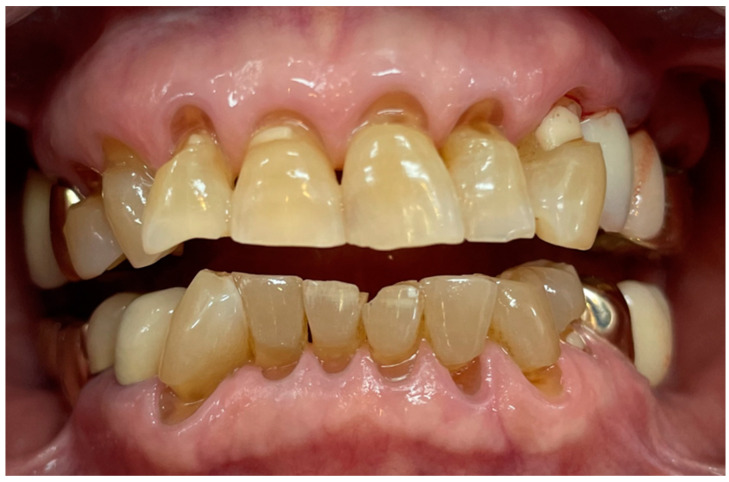
A 76-year-old female patient with multiple cervical abrasions evident due to her brushing habit.

**Figure 2 bioengineering-11-00474-f002:**
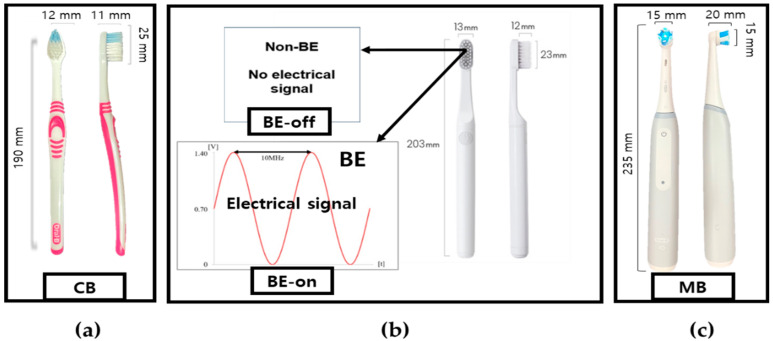
Actual figure of tested toothbrushes. (**a**) Conventional manual toothbrush (CB). (**b**) Proposed BE toothbrushes (applied non-BE: BE-off, applied BE: BE-on). (**c**) Electric–mechanical toothbrush, Oral-B iO3, (MB).

**Figure 3 bioengineering-11-00474-f003:**
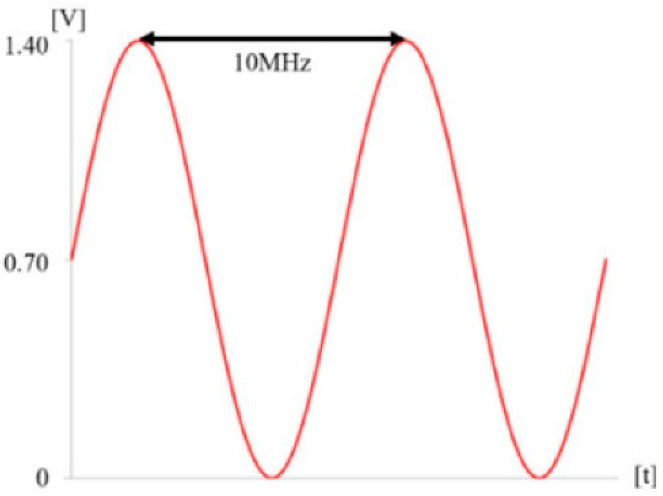
Schematic of electrical signal for BE.

**Figure 4 bioengineering-11-00474-f004:**
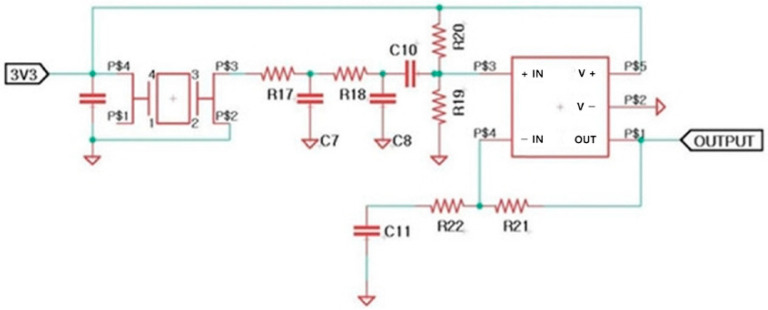
Schematic of an electronic system for generating electric signal for BE.

**Figure 5 bioengineering-11-00474-f005:**
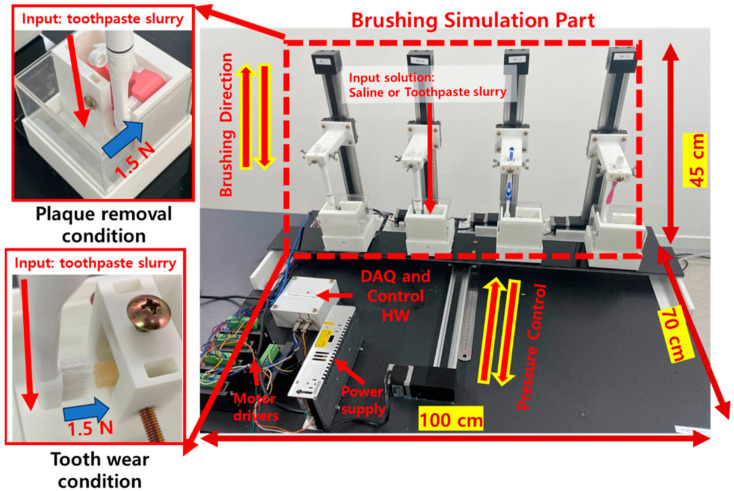
The development of the brushing simulator [21].

**Figure 6 bioengineering-11-00474-f006:**
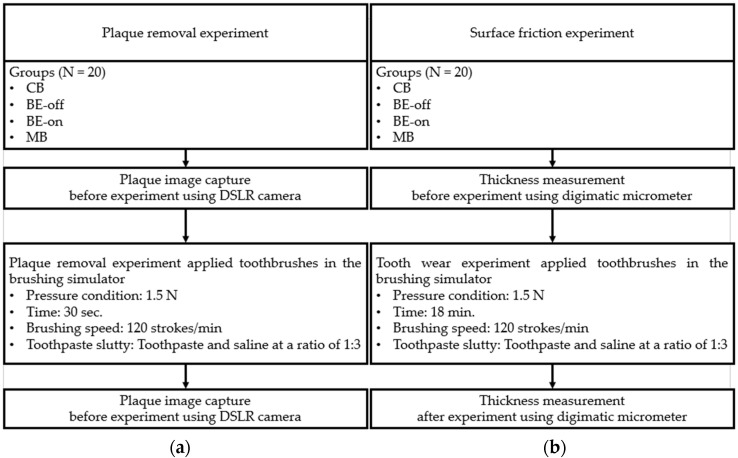
Graphic description of the plaque removal and tooth wear procedure: (**a**) plaque removal experiment (N = 20). (**b**) Wear experiment (N = 20).

**Figure 7 bioengineering-11-00474-f007:**
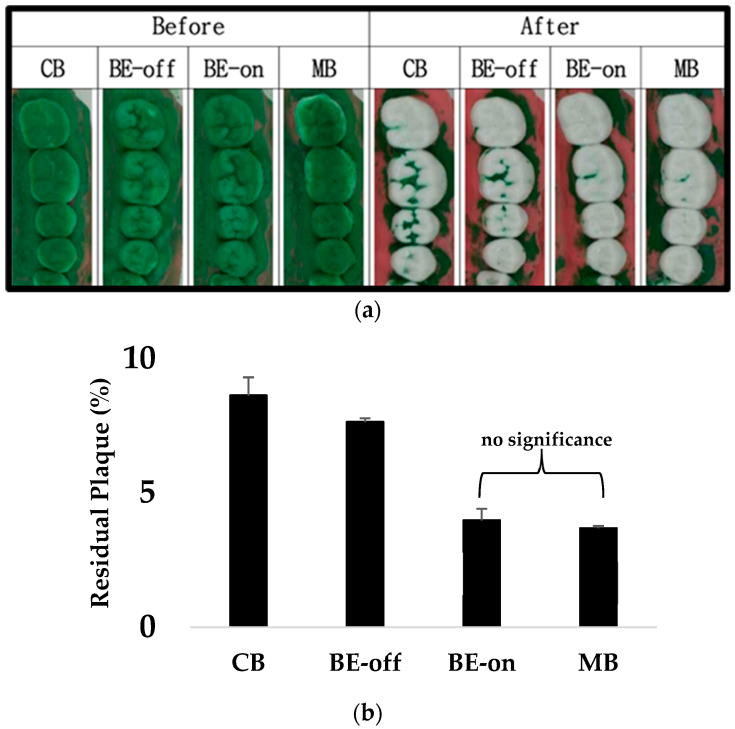
Residual plaques in plaque removal experiment: (**a**) representative image showing significant plaque reduction under BE-on and MB; (**b**) results of residual plaque (N = 5 in each condition). Results are presented as means ± SDs. * *p* < 0.05 versus CB. There was no significant difference between BE-on and MB.

**Figure 8 bioengineering-11-00474-f008:**
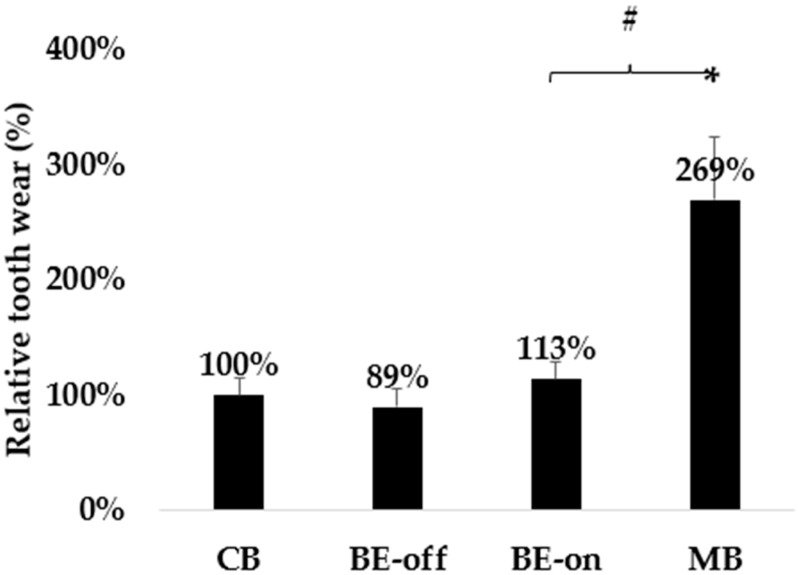
Results of relative wear in wear experiment showing significant relative wear under only MB (N = 5 in each condition). Results are presented as means ± SDs. * *p* < 0.05 versus CB. # *p* < 0.05 versus BE-on.

**Table 1 bioengineering-11-00474-t001:** Description of condition for tested toothbrushes.

Toothbrushes	Abbreviations	Note
Oral-B Ultra-fine, Oral-B Laboratories, Boston, MA, USA	CB	Typical toothbrush
Non-bioelectric effect	BE-off	non-BE
0.7 V amplitude of 10 MHz with 0.7 V offset	BE-on	Applied BE
Oral-B iO3, Oral-B Laboratories, Boston, MA, USA	MB	Electric–mechanical toothbrush

**Table 2 bioengineering-11-00474-t002:** Details of electric signal for BE.

Contents	Details	Comments
Intensity	0.7 V	Below-water electrolysis 0.82 V
Frequency	10 MHz	Effective frequency
Composition (AC:DC)	1:1	Effective biofilm treatment

**Table 3 bioengineering-11-00474-t003:** Results of the PMMA thickness change in the surface friction experiment showing significant wear under only MB (N = 5 in each condition). Results are presented as means ± SDs. * *p* < 0.05 versus CB. ^#^ *p* < 0.05 versus BE-on.

Conditions of the Toothbrush	Changes of the PMMA before and after the Surface Friction Wear Experiment (N = 5, Each Condition)
CB	7.46 ± 1.12 μm
BE-off	6.41 ± 1.23 μm
BE-on	8.33 ± 1.18 μm
MB	20.25 ± 4.02 *^,#^ μm

## Data Availability

The original contributions presented in the study are included in the article, further inquiries can be directed to the corresponding author.

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
