# Peer review of "Comparative Analysis of Plaque Removal and Wear between Electric–Mechanical and Bioelectric Toothbrushes"

_bioengineering, 2024, doi:10.3390/bioengineering11050474_

Round 1

Reviewer 1 Report

Comments and Suggestions for Authors

L49,50 Moreover, mechanical electric toothbrushes can accelerate tooth wear when used in conjunction with toothpaste [28].  Please insert Charcoal before toothpaste as regular toothpaste does not cause tooth wear.

2. Materials and Methods

2.1. Tested Toothbrushes and the Design of BE Toothbrush

L 90 The BE applied to remove biofilms is a combination of electrical signals, specifically a 0.7 V sinusoidal signal at 10 MHz with a DC offset of 0.7 V, as shown in Table 2 and Figure 3 [12,20,21].

Most readers will not understand how this works. Is it battery charged? Or mains charged? Does it affect artificial pacemakers and any other medical devices?

L 106 The current from the BE signal was determined to be safe [20,21] compared to the reported biocompatible current range [35]. What was the current? Safety issues have not been discussed.

L 93 The DC offset was set below an electrolysis threshold of 0.82 V. Our previous studies revealed that this signal does not induce electrolysis [20,21]. The authors should explain what is meant by electrolysis in the context of the mouth.

L 72,73 The BE toothbrush not only demonstrated effective plaque removal but also did not cause accelerated tooth  wear. This sentence should be in the conclusion.

L78 a mechanical electric toothbrush were … Powered toothbrush is the conventional & more widespread term used. Suggest powered replaces mechanical electric throughout.

2.4. Experiment

N=3 in both arms of the study. The sample size is very small.

Figure 6. has ‘toothpaste slutty’.  Slutty is English slang for a woman of low morals. The authors meant to use SLURRY. 

2.4.1. Plaque Removal Experiment

L 127 The typodonts were coated with cultured artificial plaques. Plastic typodonts attract more plaque than enamel.

 2.4.2. Tooth Wear Experiment

L 145 Polymethyl methacrylate (PMMA) disk resin blocks (A3; Yamahachi, Gamagori, Japan) were used as test specimens. How can the authors state this was tooth wear when PMMA was used as a proxy? PMMA will not behave the same as enamel, is much softer and will abrade much more easily.

L 151 The thickness of the samples was measured using a digital micrometer before and after the experiment. This is a very crude measurement of the wear. It should be noted that abrasion will lead to a rough surface with asperities and valleys such that the micrometer will only measure form the highest asperity across the PMA disc.

3. Results

The images in fig 7a are very dark. Please can they be made lighter?

3.2. Experiment with Tooth Wear

L 184 These results are similar to those reported previously. Place under discussion.

L 186 Microcurrents do not affect enamel microhardness [40]. In addition, the BE signals used in this study are biocompatible [35]. These sentences belong in the Discussion section.

Table 3. Results of thickness change due to tooth wear experiment in tooth wear experiment showing significant relative tooth wear under only MB.

This should read Reduction in thickness of PMMA samples & not merely change.

L 192 The elastic modulus, hardness, and wear resistance of PMMA teeth are poor compared to those of natural teeth, leading to rapid wear of PMMA teeth [41]. Therefore, tooth wear was compared using relative rather than absolute values. Now we read that the substrate was not tooth. This sentence again should be under the Discussion & not in the Results.

4. Discussion

L 225 PMMA is not only used as a tooth-filling material but is also used in various … Which restorative filling material are the authors referring to?

The reviewer knows of no permanent restorative based filling material on PMMA unless the authors confuse dental composite BIS- GMA.

Authors must discuss the limitations of this study and explain how the bioelectric current works in addition to the brushing effect. The title is misleading as tooth wear is NOT studied but wear of PMMA. The title must be changed.

Comments on the Quality of English Language

The English needs improvement. Fig 6 has 'slutty' when the term should be slurry. Slutty is slang for a loose woman. 

Author Response

Reviewer #1

#1. L49,50 Moreover, mechanical electric toothbrushes can accelerate tooth wear when used in conjunction with toothpaste [28].  Please insert Charcoal before toothpaste as regular toothpaste does not cause tooth wear.

The manuscript has been revised accordingly.

“Moreover, mechanical electric toothbrushes can accelerate tooth wear when used in conjunction with charcoal toothpaste [28].”

#2. L 90 The BE applied to remove biofilms is a combination of electrical signals, specifically a 0.7 V sinusoidal signal at 10 MHz with a DC offset of 0.7 V, as shown in Table 2 and Figure 3 [12,20,21].

Most readers will not understand how this works. Is it battery charged? Or mains charged? Does it affect artificial pacemakers and any other medical devices?

We have revised the manuscript with explanation of the power and biosafety.

Introduction

“Biofilms are composed of polysaccharides and electrically polarized bacterial cells [10,11]. Biofilms are also extremely persistent, as they are 500–5,000 times more resistant to antibiotics than biofilms in their native state [12–14]. However, external electrical stimuli can disrupt the structural integrity of biofilms and affect the bacterial metabolic state based on the principles of bioelectric effect (BE) [15, 16]. The BE has been demonstrated the biocompatibility as shown non-electrochemical condition changes due to the electric current applications as well as human clinical studies in previous [15-20]. Hence, the BE can be an additional effective method for biofilms removal in healthcare industries [17–20].

Material and Methods

“The BE toothbrush is integrated with a rechargeable battery (3.7V) to operate electronics. The BE toothbrush has been certified home appliances safety internationally including US-FDA, EU-CE, and Korea. It is safe to human application. The BE is a combination of electrical signals, specifically a 0.7 V sinusoidal signal at 10 MHz with a DC offset of 0.7 V, as shown in Table 2 and Figure 3 [12,20,21]. The BE signal frequency was selected based on a previous study [15]. The DC offset was set below an electrolysis threshold of 0.82 V. Our previous studies revealed that this signal does not induce electrolysis that does not create any chemical radicals [20,21].”

#3. L 106 The current from the BE signal was determined to be safe [20,21] compared to the reported biocompatible current range [35]. What was the current? Safety issues have not been discussed.

We have revised the manuscript with additional safety description in introduction, material and methods sections.

Introduction

“However, external electrical stimuli can disrupt the structural integrity of biofilms and affect the bacterial metabolic state based on the principles of bioelectric effect (BE) [15, 16]. The BE has been demonstrated the biocompatibility as shown non-electrochemical condition changes due to the electric current applications as well as human clinical studies in previous [15-20]. Hence, the BE can be an additional effective method for biofilms removal in healthcare industries [17–20].”

Material and Methods

“The BE toothbrush is integrated with a rechargeable battery (3.7V) to operate electronics. The BE toothbrush has been certified home appliances safety internationally including US-FDA, EU-CE, and Korea. It is safe to human application. The BE is a combination of electrical signals, specifically a 0.7 V sinusoidal signal at 10 MHz with a DC offset of 0.7 V, as shown in Table 2 and Figure 3 [12,20,21]. The BE signal frequency was selected based on a previous study [15]. The DC offset was set below an electrolysis threshold of 0.82 V. Our previous studies revealed that this signal does not induce electrolysis that does not create any chemical radicals [20,21].”

“Since the BE is characterized on the consideration of the biosafety, the current from the BE signal was determined to be safe [20,21] referred to the reported biocompatible current range, under 1000 micro-ampere [35].”

#4. L 93 The DC offset was set below an electrolysis threshold of 0.82 V. Our previous studies revealed that this signal does not induce electrolysis [20,21]. The authors should explain what is meant by electrolysis in the context of the mouth.

The manuscript has been revised accordingly.

The DC offset was set below an electrolysis threshold of 0.82 V. Our previous studies revealed that this signal does not induce electrolysis that represents no byproduct of any chemical radicals [20,21].”

#5. L 72,73 The BE toothbrush not only demonstrated effective plaque removal but also did not cause accelerated tooth  wear. This sentence should be in the conclusion.

The sentence has been deleted from the introduction.

#6. L78 a mechanical electric toothbrush were  Powered toothbrush is the conventional & more widespread term used. Suggest powered replaces mechanical electric throughout.

We agree with the reviewer’s comment.

However, we believe that the “powered toothbrush” could stand for any electrical power consumed devices.

We have revised the “mechanical electric” to “electric-mechanical” toothbrush to emphasize on the mechanical electric motor operation properties.

The manuscript has been revised accordingly.

#7. N=3 in both arms of the study. The sample size is very small.

Thank you for the questions. And we understand the comment.

In this work, we have focused on the initial validation prior to the extended human involved studies using in-vitro precision automated system.

We performed three times duplicated experiments based on the previous biological experimental studies. Three times repetition of the biological events are considered as minimal number of samples to investigate any significancy of the condition. We have followed this consideration as the first experimental study of the new technology.

Also based on the increased reproducibility of the in-vitro simulators due to the precision controls of brush position, pressure, time, and contact area, the statistical analysis demonstrates significant differences within three times duplicated experiment of each condition.

For future human involved experiments, we plan to increase significant numbers of the samples over one hundred to investigate in-vivo real world validation of the devices.

We have revised to present this plan in the manuscript (conclusion).

Based on this work, additional research is needed to further apply the BE toothbrush for plaque removal and tooth wear in real environments with human involved extended clinical studies.”

#8. Figure 6. has ‘toothpaste slutty’.  Slutty is English slang for a woman of low morals. The authors meant to use SLURRY. 

The manuscript has been revised accordingly.

#9. L 127 The typodonts were coated with cultured artificial plaques. Plastic typodonts attract more plaque than enamel.

We agree with the reviewer’s comment.

The main purpose of the plaque removal comparison test is focused on the comparative studies between the electric mechanical and bioelectric toothbrushes.

With the consistent conditions of the typodonts, we believe that the analysis of results can conclude the comparative studies. We conducted statistical analysis and concluded the result accordingly.

We have addressed this content in the manuscript.

We utilized typodonts to compare the plaque removal efficacy in the consistent condition. The typodonts were coated with cultured artificial plaques.”

#10. L 145 Polymethyl methacrylate (PMMA) disk resin blocks (A3; Yamahachi, Gamagori, Japan) were used as test specimens. How can the authors state this was tooth wear when PMMA was used as a proxy? PMMA will not behave the same as enamel, is much softer and will abrade much more easily.

We used the PMMA as a simulated tooth material based on previously reported literature. Our focused on this work is quantification comparison of wear due to the friction of the toothbrush. The PMMA can be a reference material for comparison of surface friction induced wear efficacy.

The manuscript has been revised with additional reference.

“As the teeth mimicked material, Polymethyl methacrylate (PMMA) was utilized based on current uses of dentistry due to the material composition [38]. A disk resin blocks of the PMMA (A3; Yamahachi, Gamagori, Japan) were used as test specimens.”

Reference

[38] Reis KR, Bonfante G, Pegoraro LF, Conti PC, Oliveira PC, Kaizer OB. In vitro wear resistance of three types of polymethyl methacrylate denture teeth. J Appl Oral Sci. 2008 May-Jun;16(3):176-80. 

#11. L 151 The thickness of the samples was measured using a digital micrometer before and after the experiment. This is a very crude measurement of the wear. It should be noted that abrasion will lead to a rough surface with asperities and valleys such that the micrometer will only measure form the highest asperity across the PMA disc.

Thank you for the question.

As shown in figure 5, the PMMA samples are prepared smaller area than the brush contact area to avoid non-uniform surface wearing.

Figure 5. The development of the brushing simulator.

The measurement after experiment performed 3 times repeated per sample to reflect error of measurement. The results have been done for statistical analysis (ANOVA) to investigate significance (table 3.).

We have revised the manuscript to add above information.

“The sample dimension of PMMA (10mm, 10mm, 20mm,: width, length, height)  is smaller than the contact area (13mm, 23mm,: width, length) of the brush to eliminate possibility of non-uniform wear.  The thickness of the samples was measured 3 times in each samples using a digital micrometer before and after the experiment.”

#12. The images in fig 7a are very dark. Please can they be made lighter?

The figure has been replaced.

#13. L 184 These results are similar to those reported previously. Place under discussion.

The sentence has been deleted.

#14. L 186 Microcurrents do not affect enamel microhardness [40]. In addition, the BE signals used in this study are biocompatible [35]. These sentences belong in the Discussion section.

The sentence has been deleted.

#15. Table 3. Results of thickness change due to tooth wear experiment in tooth wear experiment showing significant relative tooth wear under only MB.

This should read Reduction in thickness of PMMA samples & not merely change.

The table 3 has been revised accordingly.

Table 3. Results of the PMMA thickness change in surface friction experiment showing significant relative wear under only MB. Results are presented as means ± SDs. * p < 0.05 versus CB. # p < 0.05 versus BE-on.

Conditions of the toothbrush

Changes of the PMMA before and after

the surface friction wear experiment

CB

7.33 ± 1.53 um

BE-off

6.67 ± 1.53 um

BE-on

8.67 ± 1.53 um

MB

19.67 ± 5.51*,# um

#16. L 192 The elastic modulus, hardness, and wear resistance of PMMA teeth are poor compared to those of natural teeth, leading to rapid wear of PMMA teeth [41]. Therefore, tooth wear was compared using relative rather than absolute values. Now we read that the substrate was not tooth. This sentence again should be under the Discussion & not in the Results.

The manuscript revised accordingly.

The contents have been moved to the discussion and removed “teeth” and clearly mention about the PMMA.

“PMMA was used in the in vitro experiments to evaluate wear during brushing. PMMA can be used for wear testing despite the differences between real teeth and PMMA. The elastic modulus, hardness, and wear resistance of PMMA teeth are poor compared to those of natural teeth, leading to rapid wear of PMMA teeth [41]. Therefore, wear was compared using relative rather than absolute values. PMMA is reported as a teeth mimicked material in various dental studies [42,46].”

#17. L 225 PMMA is not only used as a tooth-filling material but is also used in various … Which restorative filling material are the authors referring to? The reviewer knows of no permanent restorative based filling material on PMMA unless the authors confuse dental composite BIS- GMA.

We have revised the manuscript to clarify the purpose of the PMMA use in this study.

“PMMA is reported as a teeth mimicked material in various dental studies [40,44].”

#18. Authors must discuss the limitations of this study and explain how the bioelectric current works in addition to the brushing effect. The title is misleading as tooth wear is NOT studied but wear of PMMA. The title must be changed.

The manuscript has been revised accordingly.

In discussion,

“This study had some limitations. First, we used constant pressure and brushing speed; thus, changes in toothbrushing patterns in real life are not reflected [49]. Second, we evaluated plaque-removal efficiency in the same typodonts and area; thus, oral structures that vary from person to person are not reflected [50]. Third, wear was evaluated using PMMA as the dental material; however, this may differ from actual teeth [43]. Based on this work, additional research is needed to further apply the BE toothbrush for plaque removal and tooth wear in real environments with human involved extended clinical studies.”

  1. Conclusions

“The BE toothbrush not only shows an effective plaque-removal rate, but also does not induce accelerated wear in comparison of the electric-mechanical device. Thus, use of the BE toothbrush removes as much plaque as possible and minimize wear. Based on these unique advantages of the BE technology, the toothbrush may be a new and more efficient alternative method especially, focused on preservation of wear with competitive plaque cleaning. For future work, we plan to demonstrate various of oral care products integrated with the BE with details of mechanism investigations.”

The title has been revised accordingly.

Comparative Analysis of Plaque Removal and Wear between Electric-Mechanical and Bioelectric Toothbrushes

Reviewer 2 Report

Comments and Suggestions for Authors

This paper provided additional evidence on the use of a BE-based toothbrush to manage plaque on teeth. While reasonably well written this reviewer has some feedback that may help strengthen the paper further. This feedback is divided into the paper sections accordingly.

Introduction

1. Please provide additional information as to what specific cells (e.g. bacteria, yeast) and other content is found in dental plaque. Are there spots on the tooth it is more likely to form? What conditions encourage its formation? 

2. There is no mention of any safety concerns in the use of BE in plaque removal. How much is known (if anything) on how BE affects surrounding tissue and cells that are not the intended target? Is BE safe for repeated application and has this been studied beyond a few applications (i.e. years of regular use)?

3. Please provide some insight into what experimental models are typically used in comparing new approaches to brushing. This content could instead be added to the Materials & Methods section.

Materials & Methods

4. Was a power of the test done to confirm a sample size of 3 (i.e. n of 3) was sufficient? Why was the sample size so small? Unless a suitable power (of >80%) can be shown, I recommend repeating the tests with at least another n of 3 to increase total n and get a stronger power.

5. For plaque formation/culture was only S. mutans used? Plaque is a multi-species entity. If S. mutans was the only species used I strongly recommend replacing the word "plaque" when discussing your results from the related test with "S. mutans-based plaque" or "mono-species plaque". Are there plans to conduct tests with multi-species plaques? 

6. Was the directionality of brushing consistent? And, the placement of the brush? Could this be another limitation to the tests conducted?

7. Why was PMMA chosen instead of a dental composite to mimic teeth in the tests?

Discussion

8. Please include some discussion on the safety of using BE. What studies have been and will be done to confirm the safety associated with using BE long term?

9. What is the cost of a BE toothbrush vs a regular toothbrush? What is the lifetime of a BE toothbrush? Is any maintenance required? What additional accessories are required to operate them?

Conclusion

10. Please refer more directly to the findings of your current paper and edit the statements to be less general. The BE brush was not better than all types in wear or of all types in plaque removal.

Comments on the Quality of English Language

It was largely acceptable, but could benefit from an additional English language edit.

Author Response

Reviewer #2

This paper provided additional evidence on the use of a BE-based toothbrush to manage plaque on teeth. While reasonably well written this reviewer has some feedback that may help strengthen the paper further. This feedback is divided into the paper sections accordingly.

Introduction

#1. Please provide additional information as to what specific cells (e.g. bacteria, yeast) and other content is found in dental plaque. Are there spots on the tooth it is more likely to form? What conditions encourage its formation? 

We have revised the introduction accordingly.

We used Streptococcus mutan- based plaque incubated in a growth media as the standard model of biofilms. Based on the experimental results, we investigated the degree of plaque removal and surface wear of the BE toothbrush in comparison of the traditional electric-mechanical device.”

#2. There is no mention of any safety concerns in the use of BE in plaque removal. How much is known (if anything) on how BE affects surrounding tissue and cells that are not the intended target? Is BE safe for repeated application and has this been studied beyond a few applications (i.e. years of regular use)?

We have revised the manuscript with additional safety description in introduction, material and methods sections.

Introduction

“However, external electrical stimuli can disrupt the structural integrity of biofilms and affect the bacterial metabolic state based on the principles of bioelectric effect (BE) [15, 16]. The BE has been demonstrated the biocompatibility as shown non-electrochemical condition changes due to the electric current applications as well as human clinical studies in previous [15-20]. Hence, the BE can be an additional effective method for biofilms removal in healthcare industries [17–20].”

Material and Methods

“The BE toothbrush is integrated with a rechargeable battery (3.7V) to operate electronics. The BE toothbrush has been certified home appliances safety internationally including US-FDA, EU-CE, and Korea. It is safe to human application. The BE is a combination of electrical signals, specifically a 0.7 V sinusoidal signal at 10 MHz with a DC offset of 0.7 V, as shown in Table 2 and Figure 3 [12,20,21]. The BE signal frequency was selected based on a previous study [15]. The DC offset was set below an electrolysis threshold of 0.82 V. Our previous studies revealed that this signal does not induce electrolysis that does not create any chemical radicals [20,21].”

“Since the BE is characterized on the consideration of the biosafety, the current from the BE signal was determined to be safe [20,21] referred to the reported biocompatible current range, under 1000 micro-ampere [35].”

Discussion

“The application of BE results in high biofilm treatment efficiency [17,18,19]. BE is caused by the propagation of electromagnetic currents, waves, and voltages [14,15,16]. The biocompatibility of the technology has been validated through no effect to in-vitro electrochemical condition changes [15, 16] and safety in-vivo human clinical studies [12, 20].”

#3. Please provide some insight into what experimental models are typically used in comparing new approaches to brushing. This content could instead be added to the Materials & Methods section.

We have revised the introduction in emphasis on the purpose of this work.

“In this study, we aimed to quantitatively compare the plaque-removal efficiency and surface wear of toothbrushes using the bioelectric effect (BE) with those of electric-mechanical toothbrushes, which have high plaque-removal efficiency but are disadvantaged in terms of tooth wear. Clinical trials are inconvenient for quantitative evaluation owing to inter-individual variability; however, in vitro simulators have been used to increase the reproducibility of the experiments including toothbrush efficacy studies [27,32–34]. We compared four types of toothbrushes (conventional toothbrushes, developmental toothbrushes with and without the BE, and electric-mechanical toothbrushes) using an in vitro simulator. We used Streptococcus mutan- based plaque incubated in a growth media as the standard model of biofilms. Based on the experimental results, we investigated the degree of plaque removal and surface wear of the BE toothbrush in comparison of the traditional electric-mechanical device.”

Materials & Methods

#4. Was a power of the test done to confirm a sample size of 3 (i.e. n of 3) was sufficient? Why was the sample size so small? Unless a suitable power (of >80%) can be shown, I recommend repeating the tests with at least another n of 3 to increase total n and get a stronger power.

Thank you for the questions. And we understand the comment.

In this work, we have focused on the initial validation prior to the extended human involved studies using in-vitro precision automated system.

We performed three times duplicated experiments based on the previous biological experimental studies. Three times repetition of the biological events are considered as minimal number of samples to investigate any significancy of the condition. We have followed this consideration as the first experimental study of the new technology.

Also based on the increased reproducibility of the in-vitro simulators due to the precision controls of brush position, pressure, time, and contact area, the statistical analysis demonstrates significant differences within three times duplicated experiment of each condition.

For future human involved experiments, we plan to increase significant numbers of the samples over one hundred to investigate in-vivo real world validation of the devices.

We have revised to present this plan in the manuscript (conclusion).

Based on this work, additional research is needed to further apply the BE toothbrush for plaque removal and tooth wear in real environments with human involved extended clinical studies.”

#5. For plaque formation/culture was only S. mutans used? Plaque is a multi-species entity. If S. mutans was the only species used I strongly recommend replacing the word "plaque" when discussing your results from the related test with "S. mutans-based plaque" or "mono-species plaque". Are there plans to conduct tests with multi-species plaques? 

We have chosen the S. mutans as the model of dental plaque. According to the literature in below, S. mutans can be a dental plaque simulated model for various of oral disease in0vitro studies.

[36] Forssten SD, Björklund M, Ouwehand AC. Streptococcus mutans, caries and simulation models. Nutrients. 2010 Mar;2(3):290-8.

We have added this reference in the text.

2.3. Plaque Culture

Streptococcus mutans (KCTC 3065, Korean Collection for Type Cultures (KCTC), Jeongeup, Republic of Korea) which can be considered as a simulated plaque condition [36], were cultured in a growth medium (LB Broth, Ambrothia Inc., Daejeon, Republic of Korea) at 37 °C for 48 h to provide sufficient time for maturation.

#6. Was the directionality of brushing consistent? And, the placement of the brush? Could this be another limitation to the tests conducted?

We have revised the manuscript accordingly with details of experimental conditions.

2.4.2. Wear Experiment

A surface wear experiment was designed to investigate the wear caused by each toothbrush. ISO/TR 14569-1 specifies the force applied to the tooth as 0.5 to 2.5 N [37]. The pressure applied to the tooth was set to 1.5 N. Additionally, the toothbrushing speed was set to 120 strokes/min [37]. As the teeth mimicked material, Polymethyl methacrylate (PMMA) was utilized based on current uses of dentistry due to the material composition [38]. A disk resin blocks of the PMMA (A3; Yamahachi, Gamagori, Japan) were used as test specimens. The specimens were kept in water at 37 ℃ for one week [37]. Typically, 28 teeth are present [39], and a brushing time of 2 min is recommended [39]. Each tooth was brushed for 4.3 s. For the convenience of calculation, assuming three sides of the tooth, this was set to 1.5 s. Teeth were brushed twice a day for 3 s per tooth. Therefore, the wear test result was calculated as approximately 18 min/year. The sample dimension of PMMA (10mm, 10mm, 20mm,: width, length, height)  is smaller than the contact area (13mm, 23mm,: width, length) of the brush to eliminate possibility of non-uniform wear.  The thickness of the samples was measured 3 times in each samples using a digital micrometer before and after the experiment.

#7. Why was PMMA chosen instead of a dental composite to mimic teeth in the tests?

We used the PMMA as a simulated tooth material based on previously reported literature. Our focused on this work is quantification comparison of wear due to the friction of the toothbrush. The PMMA can be a reference material for comparison of surface friction induced wear efficacy.

The manuscript has been revised with additional reference.

“As the teeth mimicked material, Polymethyl methacrylate (PMMA) was utilized based on current uses of dentistry due to the material composition [38]. A disk resin blocks of the PMMA (A3; Yamahachi, Gamagori, Japan) were used as test specimens.”

Reference

[38] Reis KR, Bonfante G, Pegoraro LF, Conti PC, Oliveira PC, Kaizer OB. In vitro wear resistance of three types of polymethyl methacrylate denture teeth. J Appl Oral Sci. 2008 May-Jun;16(3):176-80. 

Discussion

#8. Please include some discussion on the safety of using BE. What studies have been and will be done to confirm the safety associated with using BE long term?

We have revised the manuscript through introduction and discussion to address the biosafety.

Introduction

“Biofilms are composed of polysaccharides and electrically polarized bacterial cells [10,11]. Biofilms are also extremely persistent, as they are 500–5,000 times more resistant to antibiotics than biofilms in their native state [12–14]. However, external electrical stimuli can disrupt the structural integrity of biofilms and affect the bacterial metabolic state based on the principles of bioelectric effect (BE) [15, 16]. The BE has been demonstrated the biocompatibility as shown non-electrochemical condition changes due to the electric current applications as well as human clinical studies in previous [15-20]. Hence, the BE can be an additional effective method for biofilms removal in healthcare industries [17–20].”

Material and Methods

“The BE toothbrush is integrated with a rechargeable battery (3.7V) to operate electronics. The BE toothbrush has been certified home appliances safety internationally including US-FDA, EU-CE, and Korea. It is safe to human application. The BE is a combination of electrical signals, specifically a 0.7 V sinusoidal signal at 10 MHz with a DC offset of 0.7 V, as shown in Table 2 and Figure 3 [12,20,21]. The BE signal frequency was selected based on a previous study [15]. The DC offset was set below an electrolysis threshold of 0.82 V. Our previous studies revealed that this signal does not induce electrolysis that does not create any chemical radicals [20,21].”

Discussion

“The application of BE results in high biofilm treatment efficiency [17,18,19]. BE is caused by the propagation of electromagnetic currents, waves, and voltages [14,15,16]. The biocompatibility of the technology has been validated through no effect to in-vitro electrochemical condition changes [15, 16] and safety in-vivo human clinical studies [12, 20].”

  1. What is the cost of a BE toothbrush vs a regular toothbrush? What is the lifetime of a BE toothbrush? Is any maintenance required? What additional accessories are required to operate them?

We have added contents in discussion focused on the economic impact of the BE toothbrush in comparison of traditional electric brush.

“The BE toothbrush can be an economic tool for oral health maintenance based on the same cost and lifetime with ordinary uses compared to the traditional electric-mechanical device.”

Conclusion

  1. Please refer more directly to the findings of your current paper and edit the statements to be less general. The BE brush was not better than all types in wear or of all types in plaque removal.

We have emphasized on the advantages of the BE.

  1. Less wear of the tooth in comparison on the electric-mechanical toothbrush
  2. Competitive plaque removal in comparison on the electric-mechanical toothbrush

  1. Conclusions

The BE toothbrush not only shows an effective plaque-removal rate, but also does not induce accelerated wear in comparison of the electric-mechanical device. Thus, use of the BE toothbrush removes as much plaque as possible and minimize wear. Based on these unique advantages of the BE technology, the toothbrush may be a new and more efficient alternative method especially, focused on preservation of wear with competitive plaque cleaning. For future work, we plan to demonstrate various of oral care products integrated with the BE with details of mechanism investigations.

Round 2

Reviewer 1 Report

Comments and Suggestions for Authors

Much improved after revision.

Author Response

Dear Editor,

We appreciate on the feedback and comments which contribute to improve the manuscript significantly.

We have repeated 2 times more in each experiment and updated the results.

Details of revision are following.

Hope this version could further proceed to the next step.

Please let us know if you have any further comments.

Regards,

Young Wook Kim, Ph.D.
